# Role of Neutrophils in Anti-Tumor Activity: Characteristics and Mechanisms of Action

**DOI:** 10.3390/cancers17081298

**Published:** 2025-04-11

**Authors:** Xin Chen, Bingdi Chen, Huadong Zhao

**Affiliations:** 1Department of General Surgery, Tangdu Hospital, Air Force Medical University, Xi’an 710032, China; chenx15@mails.jlu.edu.cn; 2The Institute for Biomedical Engineering & Nano Science, Tongji University School of Medicine, Shanghai 200092, China

**Keywords:** neutrophils, anti-tumor activity, phenotypes of neutrophils, trained immunity, cytotoxic T lymphocytes, trogocytosis

## Abstract

Neutrophils are vital components of the immune system, playing multifaceted roles within the tumor microenvironment. With remarkable diversity, heterogeneity, and plasticity, they exhibit both anti-tumor and tumor-promoting activities the tumor microenvironment. While their anti-tumor effects are substantial, the full spectrum of their functions in the tumor microenvironment remains incompletely understood. This review delves into their anti-tumor mechanisms, such as reactive oxygen species -induced tumor cell destruction, cytotoxic T lymphocytes-mediated killing, cytotoxic enzymes, trogocytosis, and trained immunity. Additionally, this review also explores the therapeutic potential of neutrophil-targeted strategies in cancer treatment, offering insights into their future application as innovative anti-tumor approaches.

## 1. Neutrophils: An Overview of Functions, Development, and Polarization

Neutrophils are the most abundant leukocytes in human peripheral blood, accounting for 50–70% of white blood cells (10–25% in mice). Under homeostatic conditions, mature neutrophils are predominantly stored in the bone marrow, with only 1–2% circulating in the bloodstream. During immune responses, neutrophils are recruited to sites of infection or inflammation through the leukocyte adhesion cascade. This recruitment process is regulated by the downregulation of C-X-C chemokine receptor 4 (CXCR4) and the upregulation of CXCR1 and CXCR2 signaling pathways [1]. Once in the tissue, neutrophils rapidly execute their immune functions but have a short lifespan—approximately 7 h in humans and 8–10 h in mice [2]. As an important cell type of innate immune response, neutrophils are responsible for host defense, tissue damage, and wound healing [3]. Their anti-infective mechanisms encompass diverse strategies, including phagocytosis; degranulation of cytotoxic enzymes; formation of neutrophil extracellular traps (NETs); release of reactive oxygen species (ROS); apoptosis; expression of tumor necrosis factor-related apoptosis-inducing ligands (TRAILs), nitric oxide, and chemokines; antigen presentation; and antibody-dependent cellular cytotoxicity (ADCC) [4,5,6,7,8,9,10].

Neutrophils are derived from hematopoietic stem cells (HSCs) in the bone marrow, which differentiate into multi-potent progenitor (MPP) cells and lose their self-renewal ability. Subsequently, MPP cells differentiate into common myeloid progenitor cells (CMPs), which can continue to differentiate into granulocyte-monocyte progenitor (GMP) cells. As shown in Figure 1, GMP cells subsequently differentiate into band neutrophils and finally into mature neutrophils. GMP cells express CD34 and CD117, while promyelocytes begin to express CD33, accompanied by the disappearance of CD34 and CD117 [11]. CD33 is an antigen marker in myeloid cells and regulates cell growth and differentiation by recruiting signaling molecules. With the continuous maturation of neutrophils, band neutrophils begin to express CD15. Mature segmented human neutrophils express CD33, CD66b, and CD15. Mature mice neutrophils have markers for CD11b and lymphocyte antigen complex locus G6 (Ly6G), while KIT proto-oncogene (C-Kit) is its immature marker [12]. After being activated, neutrophils can release myeloperoxidase (MPO), neutrophil elastase (NE), cathepsin, lysozyme, defensin, alkaline phosphate enzymes, collagenase, lactoferrin, and other proteins [13].

Recent advancements in single-cell analysis have profoundly enriched our understanding of neutrophil maturation originating from GMPs. In the bone marrow of mice, GMPs differentiate into pro-neutrophils (proNeus), which represent early precursors and correspond to transcriptional clusters G0/G1 in mice and eNeP/N1 in humans, as identified in single-cell studies [14]. ProNeu cells then transition into proliferative precursors (preNeus), which play a crucial role in neutrophil expansion within the bone marrow and spleen. Transcriptionally, preNeu cells align closely with unipotent neutrophil progenitors (NePs) and intermediate precursors, including G2 clusters in mice and N2 inhuman that resemble myelocytes and metamyelocytes [14,15]. As maturation progresses, preNeu cells develop into immature neutrophils (Ly6G^int^ CXCR2^−^, immNeus) and mature neutrophils (Ly6G^high^ CXCR2^+^; matNeus). These populations serve distinct roles, with immature neutrophils exhibiting unique transcriptional profiles characterized by high secondary granule gene expression, akin to a band cell-like G3 cluster in mice (N3 cluster in humans). Mature neutrophils, distinguished by elevated expression of Matrix metalloproteinases 8 (MMP8) and C-X-C Motif chemokine Ligand 2 (CXCL2), correspond to the G4 cluster in mice (N4 cluster in humans), marking the terminal state of maturation within the bone marrow.

Neutrophils exhibit exceptional responsiveness to microenvironmental signals, enabling them to adjust their transcriptional programs and functional states, which is a property referred to as neutrophil plasticity. Similar to macrophages, neutrophils can polarize into pro-inflammatory “N1” and anti-inflammatory “N2” phenotypes, depending on the microenvironment [16,17]. However, neutrophil plasticity is manifested across multiple dimensions, including maturation status, cell density, surface marker expression, etc. Both granulocyte progenitors and mature neutrophils integrate intracellular and environmental signals to develop distinct phenotypic and functional characteristics. For instance, immature neutrophils, distinguished by their band-shaped nuclei or low-density traits, are mobilized from the bone marrow during emergency granulopoiesis and exhibit enhanced inflammatory responses and oxidative burst capacity. In contrast, aged neutrophils, characterized by upregulated CXCR4 expression and reduced CD62L surface expression, display diminished antibacterial activity but enhanced tendencies for neutrophil extracellular trap (NET) formation and immunomodulatory functions. The key factors shaping neutrophil plasticity include inflammatory mediators, metabolic cues, and tissue-specific signals. For instance, inflammatory mediators, such as lipopolysaccharide (LPS), tumor necrosis factor-α (TNF-α), and interferon-γ (IFN-γ), promote the formation of the “N1” neutrophil phenotype, characterized by enhanced antimicrobial activity and the release of pro-inflammatory mediators. Conversely, anti-inflammatory signals, including transforming growth factor-β (TGF-β), interleukin-4 (IL-4), and glucocorticoids, drive the polarization toward the “N2” neutrophil phenotype, which facilitates immune regulation and modulates inflammatory mediators.

## 2. Neutrophils and Tumors

### 2.1. Functions of Neutrophils in Tumors

In recent years, neutrophils have been shown to be involved in tumor initiation and progression, and exhibit diversity in the TME [18]. The reported effects of neutrophils in tumors are seemingly contradictory, which can be attributed to their heterogeneity of phenotype and function in tumors [19]. On the one hand, neutrophils have been shown to promote tumor growth, angiogenesis, and tumor cell metastasis while inhibiting anti-tumor T cell responses [20]. Notably, myeloid-derived suppressor cells (MDSCs), a heterogeneous cell subset, share some phenotypic characteristics with neutrophils, which can also suppress T cells and promote tumor growth and metastasis [21]. Research indicates that neutrophils in the pancreatic cancer tumor microenvironment have a lifespan of more than four times that of circulating neutrophils [22]. Additionally, they thrive in hypoxic and nutrient-poor environments, congregating near the tumor core. In these areas, they express high levels of vascular endothelial growth factor α (VEGFα), aiding in the formation of blood vessels within the tumor core, thereby supplying nutrients and fostering tumor growth. In colorectal cancer (CRC), the frequency of CD66^+^_high_ neutrophils was positively associated with CRC malignancy [23], whereas myeloperoxidase (MPO^+^) neutrophils showed the opposite trend as a good prognostic factor [24]. In CXCR2-deficient mice, administration of carcinogens failed to induce papilloma or adenoma, in which neutrophil transport is significantly impaired, and depletion of the entire neutrophil population using anti-Ly6G hinders tumorigenesis [25]. Malanchi’s group observed that radiation-stimulated neutrophils could promote lung metastasis through the Notch signaling pathway in the lung epithelium [26]. Additionally, clinical studies have observed that a high neutrophil to lymphocyte ratio (NLR) is associated with poor prognosis in patients with various cancers, including breast cancer and pancreatic ductal adenocarcinoma [27,28,29].

On the other hand, some other studies claim that neutrophils show a potent tumor suppression effect. A recent study found that successful tumor therapies acutely expanded tumor neutrophil numbers, which was positively correlated with disease outcomes in patients with lung cancer [30]. Yan et al. [31] found that polymorphonuclear neutrophils in healthy people were engaged in naturally specific cancer-killing activity (CKA), but normal primary epithelial cells were not killed while the blood neutrophils of patients with tumors and irradiated neutrophils exhibited relatively poor CKA. Coincidentally, our research team [32] found that human polymorphonuclear granulocytes exhibited high CKA when co-cultured with tumor cells, which was sharply inhibited during stress stimulation. Furthermore, Maharaj et al. [33] conducted a combined phase I/II open-label clinical trial, wherein three patients with advanced, relapsed, or refractory solid tumors were enrolled to test possible antineoplastic efficacy of human leukocyte antigen (HLA)-mismatched non-irradiated white cells (68–91% granulocytes) collected from young, healthy donors. The pathological examination revealed that the infusion of granulocytes caused extensive tumor necrosis and leukocyte infiltration. It has been reported that tumor entrained neutrophils (TENs) are significantly increased and accumulated in the lungs prior to the arrival of metastatic cells in mouse models of breast tumors [34]. TENs inhibit metastatic seeding in the lungs and acquire cytotoxic phenotypes in tumor cells, which is related to the secretion of chemokine (C-C motif) ligand 2 (CCL2) by the tumor cells [34]. The contrasting observations regarding the pro- and anti-tumor properties of neutrophils could be attributed to the differences in the donors (healthy donors and donors with different tumor types), as these neutrophils would have undergone different polarization in tumor/non-tumor microenvironments. Sagiv et al. [35] found two types of neutrophils with different densities: high-density neutrophils (HDNs) and low-density neutrophils (LDNs). Co-culture of the two types of neutrophils with tumor cells showed that HDNs are more cytotoxic and could kill tumor cells. At the same time, LDNs had an immunosuppressive function and promoted tumor cell proliferation. In tumor-free mice, more than 95% of neutrophils are HDNs, although the population of neutrophils exhibits phenotypic heterogeneity and functional diversity. Notably, recent studies have shown that the intrinsic biology of neutrophils might have a profound impact on the elimination of tumors [36], as shown in Table 1. Hirschhorn et al. found that T-cell immunotherapies engaged neutrophils to eliminate tumor antigen escape variants [37]. Linde et al. demonstrated in mouse models that neutrophils can be harnessed to induce the eradication of tumors and reduce metastatic seeding through the combined actions of tumor necrosis factor, CD40 agonist, and tumor-binding antibody [38]. In Table 1, we systematically summarize recent studies on the effects of neutrophils in tumors, offering fresh insights into their anti-tumor activities.

### 2.2. Phenotypes of Neutrophils in Tumors

Although the role of neutrophils in tumors is subject to debate, researchers increasingly acknowledge their significant contribution to tumor progression. These roles evolve in response to changes in the tumor microenvironment (TME), driven by the remarkable heterogeneity and plasticity of neutrophils. In essence, neutrophils exhibit diverse phenotypes depending on the tumor microenvironment, and these phenotypic differences further lead to varied functional impacts on tumor dynamics. Recently, the terminology ‘N1’ and ‘N2’ as pertaining to neutrophils describes the anti-tumor and pro-tumor neutrophil population in analogy to the anti-tumor ‘M1’ and pro-tumor ‘M2’ macrophages [16,17]. In 2009, Fridlender et al. [39] first proposed the boundaries of anti-tumor and tumor-promoting tumor-associated neutrophils (TANs), called ‘N1’ TANs and ‘N2’ TANs, respectively. ‘N1’ TANs exhibit a morphology that is more lobulated and hypersegmented, and they mainly achieve the purpose of suppressing tumors through neutrophil cytotoxicity and by stimulating adaptive immune cytotoxic T lymphocytes (CTLs) [52]. Moreover, ‘N1’ and ‘N2’ neutrophils are not derived entirely from the TME, as they can originate from TANs, HDNs, LDNs, or granulocyte myeloid-derived suppressor cells (G-MDSCs). This reflects their plasticity, and their heterogeneity and diversity. The G-MDSC population is heterogeneous, with immunosuppressive function in the tumor microenvironment [53]. G-MDSCs share many traits with ‘N2’ TANs, such as the promotion of tumor proliferation and angiogenesis [16,54]. G-MDSCs are derived from HSCs, while TANs can originate from either mature or immature (peripheral) cells [54].

‘N1’ TANs can express chemokines, such as chemokine (C-C motif) ligand 3 (CCL3), C-X-C motif chemokine ligand 9 (CXCL9), C-X-C motif chemokine ligand 10 (CXCL10), tumor necrosis factor-α (TNF-α), and interleukin- 12 (IL-12), and recruit and activate CD8^+^ T cells, thereby activating the anti-tumor T-cell response [55]. ‘N2’ TANs mainly exhibit tumor promotion by accelerating tumor cell proliferation, metastasis, and tumor angiogenesis [20]. ‘N2’ TANs promote tumor proliferation, metastasis, and invasion by releasing neutrophil extracellular traps (NETs), suppressing immune responses, and producing cytokines and proteases [20,56]. When stimulated by regulatory factors, such as G-CSF or TGF-β, neutrophils can transform between the ‘N1’ and ‘N2’ TAN phenotypes [39,57]. The polarization of ‘N1’ TANs is thought to be induced by type 1 interferons. The phenotypic transition from ‘N1’ to ‘N2’ TANs may indicate an antagonistic signal pathway between type 1 interferons and TGF-β. Andzinski et al. [58] found that in the absence of IFN-β, ‘N2’ TANs are the dominant neutrophil phenotypes in primary lesions and premetastatic lungs. Interferon therapy in mice altered TAN polarization towards ‘N1’ TANs [58]. TGF-β block treatment caused an increase in CD11b^+^/Ly6 TANs that are more lobulated and hypersegmented, more cytotoxic to tumor cells, and exhibit a more significant tumor-killing effect [39]. Other studies found that circulating LDNs showed less segmented nuclei and displayed impaired neutrophil function, reduced CKA, and immunosuppressive properties [35,59]. Another study [60] explored alterations in the gene expression profiles of neutrophils following TGF- β inhibition. It was shown that ‘N1’ and ‘N2’ neutrophils represent distinct subpopulations with different transcriptional signatures, such as cytoskeletal organization and antigen presentation, as well as alterations in chemokine profiles, which eventually affect their functions in tumor cells [60]. However, there are currently no suitable surface markers and no definitive method to distinguish between ‘N1’ and ‘N2’ neutrophils, and the distinction between them is mainly indicated by comparing the chemokine profiles [61], cell densities, and morphologies, as illustrated in Figure 2. N1-type neutrophils, characterized by their anti-tumor phenotype, exhibit hypersegmented nuclei and high expression levels of factors such as tumor necrosis factor-α (TNF-α), intercellular adhesion molecule-1 (ICAM-1), vascular cell adhesion molecule-1 (VCAM-1), and CCL3 [21]. These neutrophils can activate CD8^+^ T cells and demonstrate potent tumor-cell-killing capacity in vitro. Conversely, N2-type neutrophils, which are pro-tumorigenic, typically display ring-shaped nuclei and are distinguished by their ability to produce and release substantial amounts of arginase 1 (Arg1), neutrophil extracellular traps (NETs), and inducible nitric oxide synthase (iNOS) [21]. N2 neutrophils also exhibit elevated levels of CXCR4, vascular endothelial growth factor (VEGF), S100 calcium-binding proteins A8/A9 (S100A8/A9), CCL2, and CCL5. Depleting this population of cells has been shown to activate intratumoral T cells and suppress tumor growth.

In recent years, advances in single-cell transcriptomics have significantly enhanced our understanding of the heterogeneity of neutrophils within the TME [62,63]. As summarized in Table 2, studies have revealed that neutrophils exhibit diverse phenotypes across different tumor types, tissues, and species. In non-small cell lung cancer (NSCLC), TANs are classified into four distinct subpopulations [64]. TAN-1, which facilitates neutrophil activation, recruitment, and the formation of NETs, is characterized by the expression of genes such as CXCL1 and CXCL8. TAN-2, defined by the expression of MHC II-related genes, demonstrates anti-tumor immune potential. TAN-3, enriched in lipid metabolism-related genes and urokinase-type plasminogen activator (uPA), promotes tumor proliferation and migration. Lastly, TAN-4, which is associated with ribosomal function, may influence transitions in tumor endothelial cells. Neutrophils located in adjacent normal tissues (NANs) are also categorized into two subgroups, each playing distinct roles in regulating inflammation. This suggests the duality of neutrophils, with both pro-tumor and anti-tumor subsets coexisting within the TME. Additionally, analysis of large-scale samples (8766 samples from 31 solid cancers), using The Cancer Genome Atlas (TCGA), revealed tissue-specific infiltration and heterogeneity of neutrophils [65]. For instance, HLA-DR^+^CD74^+^ neutrophils are enriched in NSCLC, bladder cancer (BLCA), and ovarian cancer (OV), where they support antigen presentation and T cell recruitment. In contrast, VEGFA^+^SPP1^+^ neutrophils accumulate in gastric adenocarcinoma and renal cell carcinoma, aiding tumor angiogenesis. Interestingly, HLA-DR^+^CD74^+^ neutrophils overlap with the TAN-2 subset, reinforcing their roles in anti-tumor immunity. Certain subsets, such as IFIT1^+^ISG15^+^ neutrophils, exhibit immunosuppressive properties associated with high PD-L1 expression, further complicating tumor immune evasion mechanisms.

Finally, recent studies utilizing mouse models and patient-derived samples have uncovered the dynamic changes and functional characteristics of neutrophils during tumor progression. In a pancreatic cancer model, T3 neutrophils, defined by a dcTRAIL-R1 expression, are pro-tumorigenic, promoting tumor angiogenesis and growth [22]. In contrast, T1 and T2 neutrophils are linked to transcriptional and metabolic pathways. Upon entering tumors, neutrophils are reprogrammed, with gradual upregulation of dcTRAIL-R1 expression and a shift toward the T3 phenotype. T3 neutrophils can survive up to 135 h and exhibit elevated expression of genes associated with hypoxia, glycolysis, and angiogenesis (e.g., VEGFA, Hk2), contributing to tumor progression. In the Lewis lung cancer mouse model, researchers identified three neutrophil populations (PMN1, PMN2, and PMN3), with PMN2 enriched in PMN-MDSC-associated genes and PMN3 expressing chemokines and stress-related genes [66].

**Table 2 cancers-17-01298-t002:** Sequence clusters of neutrophils in different types of tumors.

Species	Tumor	Neutrophils Origin Tissue	Clusters	Signatures and/or Functions	Reference
Human	NSCLC	Tumor	TAN1-4;NAN1-2	TAN1: CXCL8, CXCL1, CXCL2, ICAM1, and CD44; TAN2: HLA-DRA, CD74, and HLA-DPB1; TAN3:PLIN2, PLPP, MAP1, LC3B, and PLAU; TAN4: RPL10, RPS2, RPS18, RPL3, NAN1, S100A12, PAD14, PROK2, and MMP9; NAN2: similar to NAN1 cluster, and decreased expression of S100A12, MME, and PROK2.	[64]
	NSCLC	Tumor	hN1–hN5	hN1: MMP8, MMP9, S100A8, S100A 9, and ADAM8; hN2: IFIT1, IRF7, and RSAD2; hN3: CASS4; hN4: CTSC; hN5: CCL3, CSF1, CTSB, and IRAK2.	[67]
	31 solid cancers	Tumor	10 distinct clusters	S100A12^+^; HLA-DR^+^CD7^4^; VEGFA^+^SPP1^+^; TXNIP^+^; CXCL8^+^IL1B^+^; CXCR2^+^; IFIT1^+^ISG15^+^; MMP9^+^; NFKBIZ^+^HIF1A^+^; ARG1^+^.	[65]
	Pancreatic ductal adenocarcinoma	Tumor	TNA0–TNN5	TAN-0: no cluster-specific distinctive features; TAN-1: terminally differentiated pro-tumor subpopulation; TAN-2: inflammatory subpopulation; TAN-3: transitional stage subpopulation; TAN-4: expression of interferon-stimulated genes; TAN-5: undefined subpopulation of low-quality cells.	[68]
	Melanoma	Peripheral Blood	hNeP and Cneut1-5	hNep: CD117^+^CD66b^+^CD38^+^ neutrophil progenitors; Cneut1: CD16^dim^CD62 L^bright^ band cell; Cneut2: terminally differentiated, mature neutrophils; Cneut3: CXCR4^+^CD49d^+^CD62 L^lo^ aged neutrophils; Cneut4: no specific features; Cneut5: immature neutrophils; Cneut6: CD16^dim^CD62 L^bright^ band cells.	[69]
Mouse	Pancreatic cancer model	Tumor	T1-3	T1: CD101^−^ dcTRAIL-R1^−^; T2: CD101^+^ dcTRAIL-R1^−^; T3: CD101^+/−^ dcTRAIL-R1^+^.	[22]
	Lewis lung cancer model	Tumor	PMN1-3	PMN1: account for almost 95% neutrophils in control spleen; PMN2: Ngp, Ltf, Cd177, Anxa1, MMP8, S100A8, S100A9, Cebpe, Ltb4r1, and Cybb; PMN3:CCL4, CCL3, CXCL2, CXCL3, SPP1,IL1B, NFKBIA, SOCS3, MIF, KLF6, ATF3, PTGS2, and XBP1.	[66]
	NSCLC lung cancer model	Tumor	mN1-6	mN1: MMP8, MMP9, S100A8, S100A9, and ADAM8; mN2: IFIT1, IRF7, and RSAD2; mN3: CXCL3; mN4: PALD1; mN5: CCL3, CSF1, CSTB, and IRAK2; mN6: FCNB and NGP.	[67]
	MMTV-PyMT transgenic mouse model of breast cancer	Tumor	C0, C2, C4, C5, C7, and C8	C0: CAMP17 and LY6g; C2: IL1β and Arg2; C4, C5: CEBPE and RETNLG; C7, C8: Tuba1b and Cdc20.	[70]

Overall, these studies profoundly underscore the plasticity and heterogeneity of neutrophils and their interactions with the TME, laying the theoretical foundation for developing novel cancer therapies targeting neutrophils. Given the high heterogeneity and diversity of neutrophils within tumors, it is anticipated that future research will focus on identifying anti-tumor and pro-tumor neutrophil phenotypes, aiming to develop therapeutic targets against them. This review will focus on the characteristics and mechanisms of neutrophils that exhibit anti-tumor effects and aims to explore the possibility of neutrophils as an anti-tumor treatment strategy in the future. At the same time, we will sort out the possible targets and methods of tumor treatment regimens for neutrophils and point out the main challenges faced by their clinical transformation, so as to provide a theoretical basis for promoting the application of neutrophils in tumor treatment.

## 3. Potential Mechanisms of Neutrophil-Induced Tumor-Killing Activity

### 3.1. Neutrophils and Reactive Oxygen Species-Induced Tumor Killing

Neutrophils can release ROS, including H_2_O_2_, which is a double-edged sword in that this both kills tumor cells and promotes tumor growth (Figure 3). Neutrophils from healthy donors naturally exhibit potent CKA against human cancer cells, dependent on ROS production. The addition of catalase (the enzyme catalyzing the decomposition of H_2_O_2_ into water and oxygen) significantly decreases neutrophil-induced CKA, suggesting that H_2_O_2_ produced by neutrophils is essential to this process [31,34]. In recent research, neutrophils activated by complement component C5a can eradicate tumors and reduce metastatic seeding via ROS production [38]. Recent advancements in β-glucan-induced trained immunity have shed light on its role in reprogramming neutrophils toward an anti-tumor phenotype by facilitating ROS [51]. Additionally, the suppression of TGF-β signaling has been linked to enhanced cytotoxic activity in CD11b^+^/Ly6G^+^ neutrophils, operating via an ROS-mediated pathway [39]. Elevated levels of ROS also stimulate the expression of receptors for advanced glycation end products (RAGEs) in tumor cells, further influencing tumor progression [16]. Neutrophils can recognize RAGEs and exert neutrophil cathepsin G-mediated cytotoxicity, independent of its proteolytic activity [16,71]. Further, the cytotoxic effect of neutrophils mediated by reactive oxygen species (ROS) has been found to rely on the expression of transient receptor potential melastatin 2 (TRPM2) on the tumor surface. This dependence underscores the interplay between ROS and TRPM2 in facilitating tumor cell death. TRPM2 is an H_2_O_2_-dependent Ca^2+^ channel, which causes Ca^2+^ to flow into the cell after being activated, leading to the apoptotic Ca^2+^-dependent cascade [40,72].

It is well known that ROS present a double-edged sword in tumorigenesis and progression. While neutrophil-derived ROS contribute to tumor growth and immunosuppression, and their dual role highlights their complexity within the TME. Tumor-derived granulocyte-macrophage colony-stimulating factor (GM-CSF) induces the upregulation of NOX2 and ROS release from granulocytes, thereby suppressing T-cell activity [73]. Additionally, studies have revealed that immature c-Kit^+^ neutrophils, relying on oxidative mitochondrial metabolism, play a critical role in mediating T-cell suppression [41].

### 3.2. Neutrophils and Cytotoxic T Lymphocytes-Induced Tumor Killing

Neutrophils acting as antigen-presenting cells (APCs) have the ability to directly stimulate the activation of CTLs [65], playing a role in tumor killing (Figure 4). Mice neutrophils derived from the bone marrow of mice with early-stage cancer exhibit minimal immunosuppressive activity. In contrast, those from mice with late-stage cancer display pronounced immunosuppressive effects by inhibiting the antigen-specific proliferation of CD8^+^ T cells [42]. Furthermore, blocking transforming growth factor-β (TGF-β) signaling in mice leads to a substantial influx of CD11b^+^/Ly6G^+^ neutrophils, which significantly decelerates tumor growth through CD8^+^ T-cell activation [39]. During TGF-β inhibitor therapy, the depletion of neutrophils results in a marked reduction in CD8^+^ T-cell activation [39]. Recently, it has been reported that neutrophil-driven resistance to primary 3-MCA sarcomagenesis is contingent upon CD4^−^CD8^−^ unconventional α β T cells (UTCαβ) by producing interferon-γ (IFN-γ) [43]. Neutrophil-dependent interleukin 12 (IL-12) production is essential for IFN-γ expression in UTCαβ [43]. Additionally, active neutrophil elastase (ELANE) has been shown to suppress primary tumor growth and generate a CD8^+^ T-cell-mediated abscopal effect, targeting distant metastases. ELANE exerts its anti-tumor effects by promoting the apoptosis of cancer cells, while simultaneously increasing the presence of tumor dendritic cells (DCs), CD8^+^ T cells, and CD8^+^ T effector cells (CD8^+^ Teffs) [50].

However, several studies have also proven that neutrophils have immunosuppressive function, which promotes tumor growth and development. Khanna et al. [73] found that an increase in the immunosuppressive CD11b^+^CD15^+^HLADR^−^ human granulocytes in patients with mesothelioma significantly hindered T-cell proliferation and activation [73]. These granulocytes exhibit low-density and potent inhibition of T-cell proliferation by producing arginase-1 and ROS [59]. Furthermore, tumor-elicited oxidative neutrophils have been known to exhibit an immature c-Kit+ phenotype and are able to suppress T cells under conditions of restricted glucose utilization. Consistent with prior studies, peripheral neutrophils from patients with cancer demonstrate increased levels of immaturity, elevated mitochondrial content, and enhanced reliance on oxidative phosphorylation [41].

### 3.3. Neutrophils and Trogocytosis

Studies have shown that neutrophils inherently lack the ability to phagocytose cancer cells. However, neutrophils equipped with Fc receptors can link with IgG or IgA antibodies present on the surfaces of tumor cells, thereby exerting antibody-dependent cellular cytotoxicity (ADCC) to eliminate tumors. This mechanism of neutrophil-driven cancer cell destruction is non-apoptotic and operates independently of both granule exocytosis and the NADPH oxidase pathway in phagocytes.

Research has reported that neutrophils inherently lack the ability to phagocytose tumor cells [74]. However, neutrophils equipped with Fc receptors can interact with IgG or IgA antibodies on the surfaces of tumor cells, facilitating antibody-dependent cellular cytotoxicity (ADCC) to eradicate tumors [44]. This mechanism of neutrophil-driven cancer cell destruction is non-apoptotic and operates independently of both granule exocytosis and the NADPH oxidase pathway in phagocytes. Further research found that the interaction between neutrophils and breast cancer cells has been shown to result in trogocytosis, whereby neutrophils acquire fragments of the target cell membrane (Figure 5). The killing mechanism, known as trogoptosis, executed by neutrophils, relies on MAC-1 (CD11b/CD18) integrin-dependent conjugate formation. This process is inhibited by the CD47-SIRPα checkpoint blockade. Antibody-based neutrophil tumor treatment remains to be further studied. All antibody types used in clinical practice are aimed at adaptive immunity, and the treatment strategy involving neutrophils has not been considered.

### 3.4. Neutrophils and Cytotoxic Enzymes

#### 3.4.1. Neutrophil Extracellular Traps

NETs have a DNA skeleton and inlaid mesh structure and are equipped with active extracellular proteins, such as myeloperoxidase, NE, MMP-), etc. In the TME, tumor cells enhance the inflammatory properties of neutrophils and activate them to release NETs [75]. Interleukin-8 (IL-8) and RAGEs produced by tumor cells can recruit neutrophils to form NETs [45,46]. Meanwhile, NETs actively recruit additional neutrophils, thereby aiding tumor cells in evading immune responses. Amyloid β, produced by cancer-associated fibroblasts (CAFs) from melanoma and pancreatic cancer, has been shown to stimulate the formation of NETs through a mechanism dependent on ROS production [47]. In turn, NETs also play a role in promoting tumor growth during tumor progression [45,76]. The NE in NETs can promote tumor cell proliferation by activating the NF-κB signaling pathway. Isolated NETs have the ability to promote the malignant transformation of breast cancer cells by changing the phenotype of CD44^−^/CD24^+^ to CD44^+^/CD24^−^, which is related to tumor migration and metastasis [48]. In lung adenocarcinoma cells, the capture of tumor cells by NETs is achieved by the expression of β1-integrin on tumor cells [77]. Moreover, the receptor for advanced glycation end products (RAGEs) expressed on neutrophils plays a pivotal role in tumor progression [16]. High-mobility group box 1 (HMGB1), a key component associated with neutrophil extracellular traps (NETs), interacts with RAGEs on tumor cells, triggering NF-κB signaling activation in glioma cells [45].

#### 3.4.2. Neutrophil Elastase

NE, a type of serine protease, is synthesized in promyelocytes and retained within neutrophils in an inactive state [78]. NE can activate and up-regulate tumor-related signaling pathways, such as EGFR/MEK/ERK and PI3K signaling pathways [49,79]. NE promotes tumor cell proliferation by blocking tumor apoptosis mediated by tumor necrosis factor-α (TNF-α) and stimulates the release of vascular endothelial growth factor (VEGF) to promote tumor angiogenesis [80]. However, Cui et al. [50] reported that catalytically active neutrophil elastase (ELANE) selectively eradicates various kinds of tumor cells while sparing non-tumor cells, thereby attenuating tumorigenesis. Further, ELANE facilitates the proteolytic liberation of the CD95 death domain, which interacts with histone H1 isoforms to specifically destroy tumor cells. Additionally, ELANE has been found to suppress primary tumor growth effectively [50].

### 3.5. Neutrophils and Apoptosis

Tumor necrosis factor-related apoptosis-inducing ligands (TRAILs) selectively induce apoptosis in tumor cells without killing normal cells by interacting with the death receptors 4/5 (DR4/5) [81]. As shown in Figure 6, TRAILs bind to DR4 and DR5 and recruit intracellular linker molecules that bind to caspase enzymes and promote apoptosis in tumor cells [82]. It was found that TRAIL expression is more than 10-fold higher in neutrophils than in peripheral blood mononuclear cells (PBMCs) [83]. Neutrophils express TRAILs and release them into the culture medium to induce apoptosis of tumor cells. INF-γ enhances the expression of TRAILs in neutrophils and promotes tumor cell apoptosis [83]. Further evidence of the importance of neutrophils in this setting comes from other works showing that TRAILs from BCG/IL-17 (BCG: Bacillus Calmette-Guérin) stimulated neutrophil-potentiated tumoricidal activity [84,85]. However, the precise mechanism by which neutrophils modulate TRAILs within the tumor microenvironment remains elusive, necessitating further comprehensive investigation.

### 3.6. Neutrophils and Trained Immunity

Trained immunity is an emerging concept that describes the innate immune system’s capacity to develop immune memory, offering prolonged protection against foreign invaders [86,87]. Previously, immune memory was thought to be a feature exclusive to adaptive immunity. However, current research has proved that innate immune cells can acquire adaptive characteristics following sufficient priming [86,88]. BCG vaccination has been shown to establish a lasting transcriptional program tied to myeloid cell development and induce trained immunity through the hematopoietic progenitor compartment in healthy individuals [89]. Additionally, Suttmann highlighted the crucial role of neutrophil granulocytes in the effective application of BCG immunotherapy for bladder cancer [90]. It is believed that neutrophils activated by BCG serve as a major source of chemokines, including IL-8 and macrophage inflammatory protein-1A. Due to limitations in the understanding of trained immunity previously, it was not proposed whether BCG immunotherapy of tumors can exert anti-tumor effects by inducing trained immunity in neutrophils. In 2018, Mitroulis and colleagues demonstrated that administering β-glucan to mice promoted the proliferation of myeloid progenitors [91]. This boost in myelopoiesis, driven by trained immunity, enhanced the response to secondary lipopolysaccharide (LPS) exposure and conferred protection against immunosuppression induced by chemotherapy. Further, Chavakis et al. demonstrated that β-glucan induces an anti-tumor phenotype in neutrophils by rewiring granulopoiesis and reprogramming neutrophil function. This trained granulopoiesis, driven by β-glucan, was shown to be transmissible through bone marrow transplantation [51]. Pre-treatment with β-glucan reduced tumor growth by reprogramming neutrophils at transcriptomic and epigenetic levels. Furthermore, the adaptive transfer of these re-programmed neutrophils to naive recipients effectively suppressed tumor growth.

However, how the characteristics of neutrophil recognition of tumor cell patterns and phenotypic changes result in trained immunity remains unclear. Furthermore, whether neutrophils are associated with tumor immunotherapy remains to be elucidated, and this is an additional direction of future research.

## 4. Neutrophils as a New Strategy in Tumor Treatment

The potential targets and strategies for tumor treatment involving neutrophils have been thoroughly explored. Neutrophils could achieve tumor therapy in various strategies, as shown in Figure 7, including neutrophil-blocking therapy, tumor-associated neutrophil-remodeling therapy, neutrophil immune checkpoints, adoptive neutrophil therapy, and tumor vaccine therapy targeting neutrophils.

(1)**Neutrophil-blocking Therapy**: In the past, numerous studies have suggested that neutrophils promote tumor initiation, progression, and metastasis, making neutrophil-blocking therapy a targeted tumor strategy [92]. Chemokine (C-X-C motif) receptor 1 (CXCR1) and chemokine (C-X-C motif) receptor 2 (CXCR2), expressed on neutrophils, guide integrin activation, and neutrophil recruitment. Blocking CXCR1/CXCR2 signaling has been considered a potent strategy for neutrophil-targeted cancer therapy [93,94]. A phase 1b trial in patients with metastatic breast cancer found that combining reparixin with paclitaxel was safe [95]. However, the primary endpoint of extending progression-free survival was not met in the phase 2 trial [96]. There are no CXCR2-targeting drugs that have been approved for cancer treatment now; other CXCR2 antagonists are in various stages of clinical development [97]. NETs are seen as promising emerging targets in tumor therapy due to their ability to induce tumor cell metastasis [98]. Treatment could involve inhibiting the formation and/or activation of NETs in tumor tissues. However, clinical trials have not yet identified the best way to target NETs, possibly due to the lack of a good biomarker for NET treatment response.(2)**Tumor-associated Neutrophil-remodeling Therapy**: Due to the diversity and plasticity of neutrophils in tumors, several pathways can remodel tumor microenvironment neutrophils to achieve anti-tumor phenotypes. Inhibiting the TGF-β signaling pathway can induce anti-tumor TAN1 phenotypes and other anti-tumor immune cells. Neutralizing TGF-β increases neutrophil–chemoattractant production, leading to the recruitment and activation of anti-tumor neutrophil subsets. Currently, various clinical-stage TGF-β-targeted drugs include monoclonal antibodies, TGF-β R fusion proteins, antisense oligonucleotides, and TGF-β R kinase inhibitors [99]. The polarization of N1 TANs can also be induced by type 1 interferons. Interferon-α (IFN-α) was approved by the FDA in 1986 for treating hairy cell leukemia, malignant melanoma, and other cancers [100]. However, cytokines have complex mechanisms in the body and can easily cause adverse reactions after anti-tumor immune therapy administration. Future research into nano-mechanisms may improve cytokine therapy. Moreover, more studies are needed to explore the molecular mechanisms of neutrophil remodeling in the TME, providing more molecular targets and a theoretical basis for tumor-associated neutrophil-remodeling therapy.(3)**Neutrophil Immune Checkpoints Inhibitors**: Targeting programmed death 1 (PD1)/programmed cell death ligand 1 (PDL1) has proven effective in immunotherapy for various cancers. Studies have found that TAN2 also expresses high levels of PD-L1 [101], which can be effectively targeted. Additionally, besides TAN2, there are many G-MDSCs in the TME that exhibit immunosuppressive functions. Future research should explore more immune checkpoints for such cells.(4)**Adoptive Neutrophil Therapy**: Maharaj et al. [33] conducted a combined phase I/II open-label clinical trial with three patients with advanced, relapsed, or refractory solid tumors to test the antineoplastic efficacy of HLA-mismatched non-irradiated white cells (68–91% granulocytes) from young, healthy donors. Although neutrophil infusion did not alter patient outcomes, pathological examination revealed extensive tumor necrosis and leukocyte infiltration caused by granulocyte infusion. This may be due to the rapid remodeling of neutrophils by the tumor microenvironment post-infusion. LifT BioSciences has since identified immunomodulatory alpha neutrophils (IMANs) that differentiate into N1 neutrophils after infusion, exhibiting selective cancer cytotoxicity, granules, and cationic peptides with a positive net cell charge [102]. Preclinical data shows IMANs demonstrate stronger tumor tissue infiltration and effective pancreatic tumor killing compared to adoptive T-cell therapy. Chimeric antigen receptor (CAR) neutrophils present a more promising research direction, targeting tumors precisely. The company engineered human epidermal growth factor receptor 2 (HER2) CAR on induced pluripotent stem cell-derived IMANs, enhancing its cancer-killing ability fourfold. Both neutrophil therapies are currently applied for clinical trials, and we look forward to their clinical data. Despite their selection or modification, these immunomodulatory alpha neutrophils still face the challenge of being remodeled by the tumor microenvironment after entering tumors. Future research should focus on neutrophil phenotypes and the regulatory factors for phenotype conversion, providing a detailed theoretical basis for adoptive neutrophil therapy. Additionally, neutrophil research should focus more on the molecular mechanisms of neutrophil recognition of tumor cells and their anti-tumor effects to design more reasonable CAR neutrophils.(5)**Tumor Vaccine Therapy Targeting Neutrophils**: Recent studies have confirmed that HLA-DR^+^CD74^+^ antigen-presenting neutrophils can induce T-cell responses in tumor tissues, forming a “hot tumor” microenvironment favorable for anti-tumor immunity [65]. This provides a new research direction for tumor vaccine design, targeting HLA-DR^+^CD74^+^ antigen-presenting neutrophils to turn “cold” tumors into “hot” tumors, remodel the TME, and enhance the efficacy of tumor vaccine immunotherapy. Additionally, research has found that neutrophils can exert anti-tumor effects through trained immunity [51], suggesting a future direction for tumor vaccine design using this approach. However, how neutrophil recognition of tumor cell patterns and phenotypic changes lead to trained immunity remains unclear. Furthermore, whether neutrophils are involved in trained tumor immunotherapy is another future research direction.(6)**Drug/Gene Delivery Based on Neutrophils:** Neutrophils, as critical participants in the tumor microenvironment, respond to tumor signals via chemokines, such as CXCL8 and CXCL5, accumulating in large numbers at tumor sites [103]. This behavior provides a novel direction for drug and gene delivery research. Neutrophil-based delivery systems offer distinct advantages, including efficient phagocytic ability, specific chemotaxis, and rapid responsiveness [104,105]. The delivery methods based on neutrophils can be categorized as follows: (1) Ex vivo drug/gene loading and reinfusion: Neutrophils are cultured ex vivo to internalize nanodrugs before being reinfused into the body. For instance, Li et al. [106] developed a method utilizing bone marrow-tropic neutrophils (BMTNs) to deliver drugs to bone marrow, significantly improving therapeutic efficacy. (2) Engineered neutrophils: Bao et al. [107] designed CLTX-CAR structures containing T-cell or neutrophil-specific signaling domains and loaded chemotherapy drugs. This approach enabled CAR-neutrophils to deliver and release tumor-responsive nanodrugs non-invasively in gliomas, maintaining their anti-tumor N1 phenotype. (3) Biomimetic nanoparticles: Cell-membrane-coated nanoparticles (CNPs) are fabricated by extracting fragments from parental cell membranes, inheriting their characteristics. For example, Wang et al. [108] developed neutrophils and macrophage membrane coated-PLGA/RAPA nanoparticles loaded with rapamycin, which could autonomously cross the blood–brain barrier. These biomimetic nanoparticles combined macrophage-stimulated self-recurrence and neutrophil-mediated inflammatory chemotaxis, effectively enhancing glioma treatment. These studies reveal the potential of neutrophil-based drug delivery systems for precision cancer therapy.

## 5. Perspectives

Neutrophils are vital components in the immune system, exhibiting diverse roles in tumor elimination and immune modulation. Emerging studies reveal neutrophils’ antigen-presenting functions, suggesting their potential in tumor vaccine development targeting the tumor microenvironment. Trained immunity, induced by agents like β-glucan, reprograms neutrophils towards anti-tumor phenotypes, offering promising therapeutic avenues. Similarly, trogocytosis and neutrophil elastase (ELANE) have demonstrated tumoricidal effects, warranting further exploration of their molecular mechanisms and safety profiles. Despite these anti-tumor attributes, neutrophils display heterogeneity and plasticity within the tumor microenvironment (TME), occasionally adopting pro-tumor phenotypes. Their polarization into tumor-suppressing N1 neutrophils or tumor-promoting N2 neutrophils is influenced by factors like interferons and regulatory cytokines (e.g., G-CSF, TGF-β). Understanding these processes and identifying molecular drivers of polarization remains a critical research direction. Recent advancements in single-cell transcriptomics have significantly deepened our understanding of neutrophil heterogeneity within the tumor microenvironment (TME). As summarized in Table 2, studies reveal that neutrophils exhibit diverse phenotypes across tumor types, tissues, and species These findings emphasize the complexity of neutrophils in the TME, with their phenotypes and functions shaped by diverse regulatory factors. Understanding this diversity is critical for advancing neutrophil-targeted therapies, including immune modulation, tumor vaccines, and adoptive neutrophil therapies. Future research must focus on unraveling the mechanisms of neutrophil reprogramming and polarization to fully harness their therapeutic potential.

Neutrophils hold promise for therapeutic applications, including neutrophil-blocking therapy, tumor-associated neutrophil-remodeling therapy, neutrophil immune checkpoints inhibitors, adoptive neutrophil therapy, tumor vaccine therapy targeting neutrophils, and drug/gene delivery based on neutrophils. While CXCR2 inhibitors and NET-targeting therapies have shown potential, challenges, such as TME-induced plasticity and reliable biomarkers, persist [96]. Adoptive therapies like IMANs and CAR-IMANs are advancing yet require a deeper investigation into neutrophil–tumor recognition mechanisms for improved efficacy. In 2023, LIfT BioSciences introduced two adoptive neutrophil therapies, IMANs and CAR-IMANs [102]. IMANs exhibit enhanced infiltration in tumor tissues and can effectively kill pancreatic-cancer-like tumors, while CAR-IMANs precisely target tumor tissues and increase IMANs’ cancer-killing ability fourfold. Both therapies lack publicly available comprehensive data, and we eagerly await their clinical data results. Despite the potential of these adoptive neutrophil therapies, the high plasticity of neutrophils poses a challenge, as they may become ineffective due to remodeling by the tumor microenvironment after entering the tumor tissue. This is a common issue faced by adoptive immune cells. For instance, T cells can become exhausted within 6–12 h after contacting tumors, NK cells lose anti-tumor activity within 24 h, and macrophages are subverted by tumors within 48 h to promote tumor growth [109,110]. Neutrophils can be reprogrammed by tumors into a long-lifespan, pro-cancer subtype within one day [22]. Therefore, future research should focus on studying neutrophil phenotypes and regulatory factors for phenotype conversion to provide a more detailed theoretical basis for adoptive neutrophil therapy.

## 6. Conclusions

Neutrophils are vital components in the immune system, exhibiting diverse roles in tumor elimination and immune modulation. This review highlights their anti-tumor mechanisms, such as ROS-induced cytotoxicity, CTL activation, trogocytosis, cytotoxic enzymes, TRAIL-induced apoptosis, and trained immunity. The potential targets and strategies for tumor treatment involving neutrophils have been thoroughly explored, including neutrophil-blocking therapy, tumor-associated neutrophil-remodeling therapy, neutrophil immune checkpoints inhibitors, adoptive neutrophil therapy, tumor vaccine therapy targeting neutrophils, and drug/gene delivery based on neutrophils.

Taken together, the diversity and plasticity of neutrophils underlie the dual functions of neutrophils in the occurrence and development of tumors. Changes in the tumor microenvironment result in different neutrophil polarizations, thereby leading to neutrophils playing different roles in tumor tissues. The literature reviewed herein also confirms that neutrophils exhibit great potential for anti-tumor treatments and the possibility of the elimination of body tumors through multiple channels. While their anti-tumor potential is substantial, further studies are essential to unravel their mechanisms and optimize them for effective cancer treatment strategies.

## Figures and Tables

**Figure 1 cancers-17-01298-f001:**
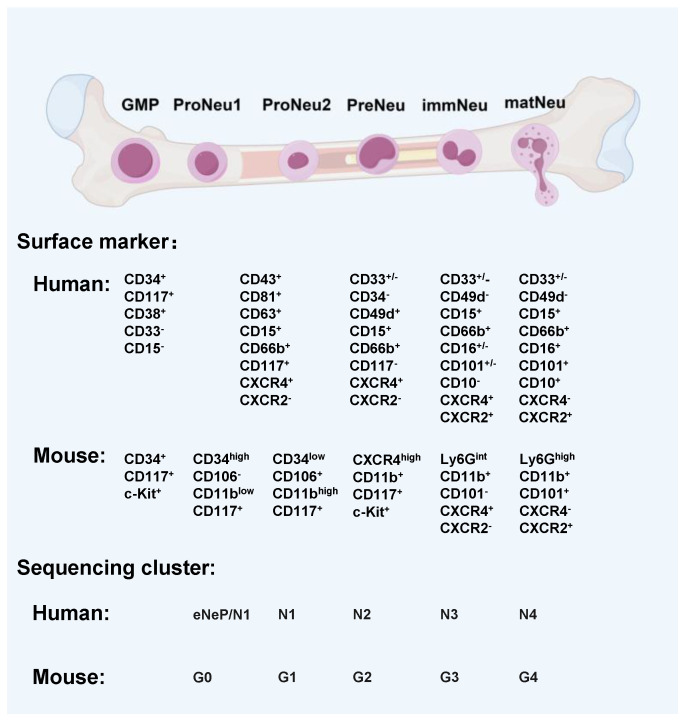
Surface markers and sequencing cluster of human and mouse neutrophils during maturation.

**Figure 2 cancers-17-01298-f002:**
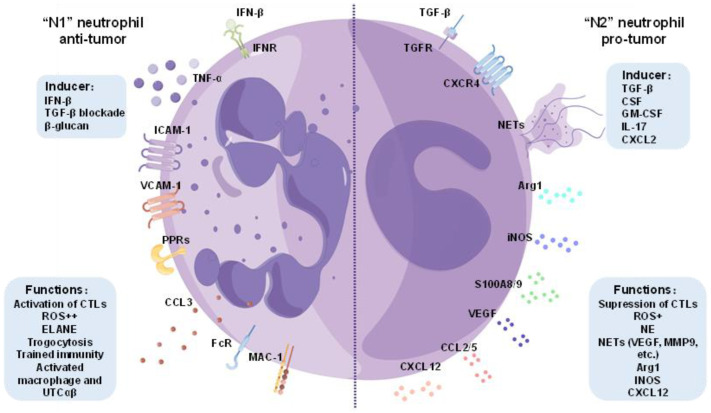
‘N1’ versus ‘N2’ neutrophils. ‘N1’ and ‘N2’ neutrophils describe the anti-tumor and pro-tumor neutrophil populations, respectively. ‘N1’ neutrophils might be related to type 1 IFN, TGF-β blockade, and β-glucan, while ‘N2’ neutrophils could be induced by G-CSF, GM-CSF, IL-17, CXCL2, and TGF-β.

**Figure 3 cancers-17-01298-f003:**
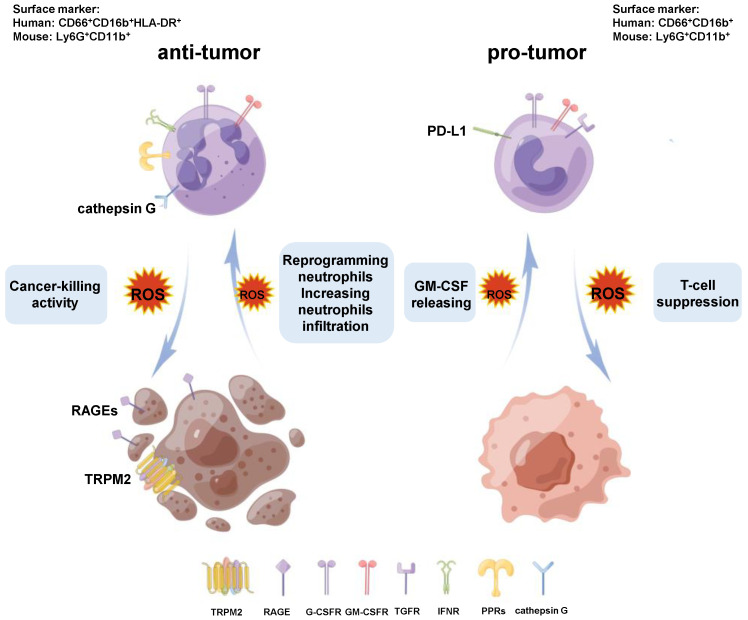
Neutrophils and ROS-induced tumor killing. ROS generated by neutrophils are a double-edged sword where tumors are concerned. Neutrophils can produce ROS to directly kill tumor cells by the interaction of cathepsin G with tumor RAGE. Moreover, the ROS produced by the neutrophils can kill tumor cells by a mechanism that depends on TRPM2 expression. Neutrophil-sourced ROS also have the ability to cause T-cell suppression.

**Figure 4 cancers-17-01298-f004:**
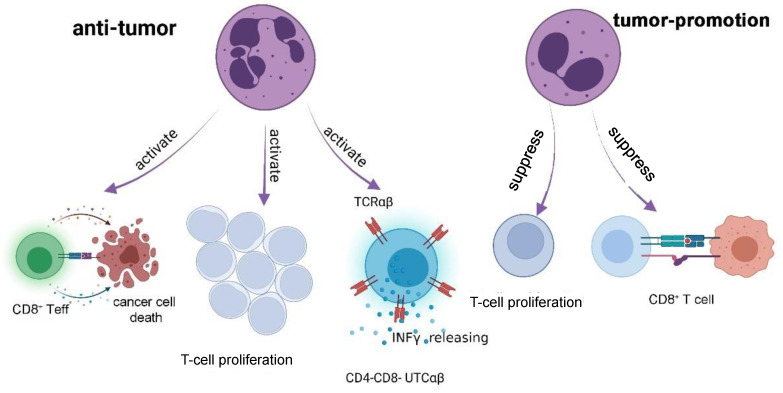
Neutrophil- and CTL-induced tumor killing. Neutrophils play a role in tumor killing by activating CTLs and CD4^−^CD8^−^ UTCαβ. The immunosuppression function of neutrophils in tumors relies on suppressing T-cell proliferation and inhibiting T-cell activation.

**Figure 5 cancers-17-01298-f005:**
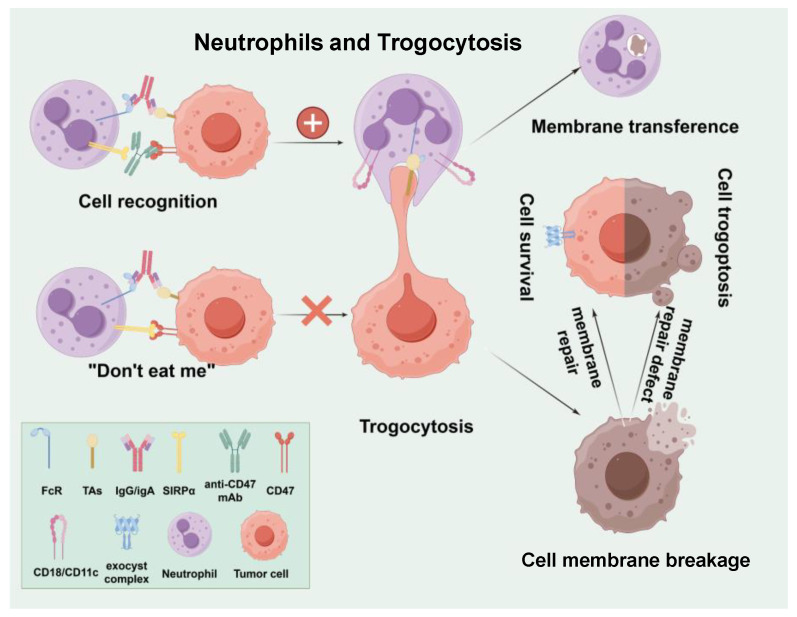
Neutrophils and trogocytosis. Neutrophils eliminate tumor cells through trogocytosis, a process reliant on CD11b/CD18 integrin-dependent conjugate formation. This mechanism can be suppressed by the CD47-SIRPα checkpoint blockade, which is crucial for immune evasion by tumor cells.

**Figure 6 cancers-17-01298-f006:**
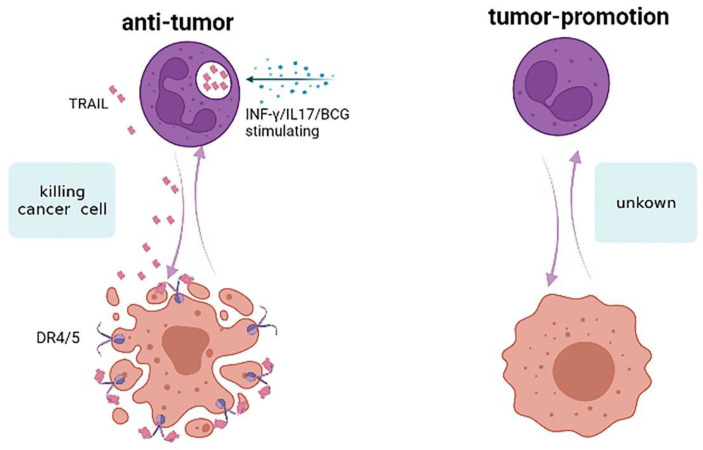
Neutrophils and TRAILs. Neutrophils express and release TRAILs, which bind to tumor DR4/5 to induce apoptosis of tumor cells. TRAILs from INF-γ/BCG/IL-17 stimulate neutrophil-potentiated tumoricidal activity.

**Figure 7 cancers-17-01298-f007:**
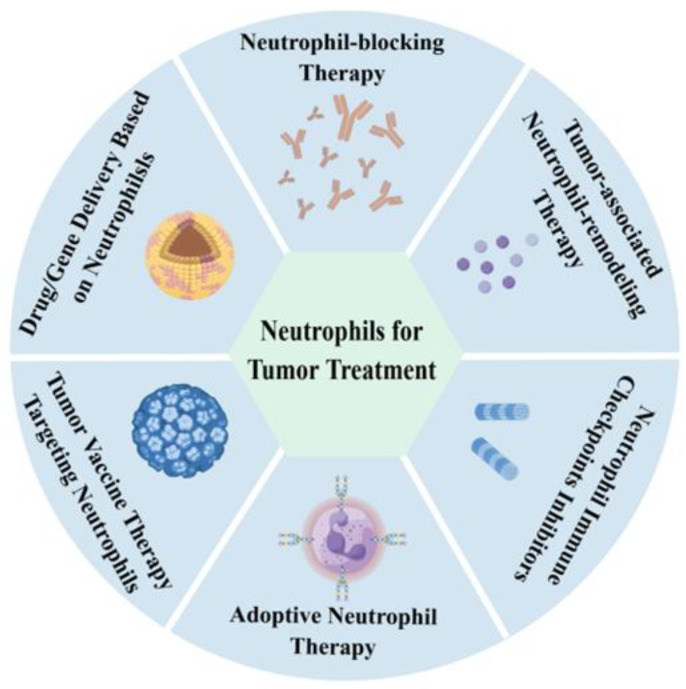
Schematic diagram of neutrophils for tumor treatment.

**Table 1 cancers-17-01298-t001:** Effects of neutrophils in different types of tumors.

Tumor Source	Neutrophil Source	Effect of Neutrophils in Tumor	Potential Mechanism	Reference
intestinal adenocarcinoma	neutrophils in spontaneous and inflammation-driven neoplasia (mice)	tumor promotion	CXCR2-MPO^+^	[24]
lung metastasis	radiation-stimulated neutrophils (mice)	tumor promotion	degranulation	[26]
ovarian cancer, cervical cancer, lung cancer (cell line)	discrete neutrophils (healthy human donors)	tumor suppression	ROS-induced tumor killing	[31]
lung cancer	discrete neutrophils (healthy human donors)	tumor suppression	ROS-induced tumor killing	[32]
breast cancer	discrete high-density neutrophils (lung cancer mice)	tumor suppression	ROS-induced tumor killing	[35]
breast cancer	discrete low-density neutrophil (lung cancer mice)	no significant tumor suppression	none	[35]
mesothelioma	TGF-β blockade stimulated neutrophil (mice)	tumor suppression	ROS-induced tumor killing, CTL-induced tumor killing	[39]
breast cancer, Lewis lung carcinoma	discrete high-density neutrophils (mice)	tumor suppression	ROS-induced tumor killing (RAGE/cathepsin G-mediated cytotoxicity)	[16]
breast cancer, Lewis lung carcinoma	neutrophils (mice)	tumor suppression	ROS-induced tumor killing	[40]
breast cancer	neutrophils (mice)	tumor promotion	oxidative mitochondrial metabolism, T-cell suppression	[41]
Lewis lung carcinoma	discrete neutrophils (mice with early-stage cancer)	tumor suppression	CTL-induced tumor killing	[42]
Lewis lung carcinoma	discrete neutrophils (mice with late-stage cancer)	tumor promotion	T-cell suppression	[42]
3-MCA sarcomagenesis	neutrophils (mice)	tumor suppression	UTCαβ induced tumor suppression	[43]
breast cancer	discrete neutrophils (unknown)	tumor suppression	trogocytosis	[44]
glioma	neutrophils (mice)	tumor promotion	NETs produced by neutrophils	[45]
pancreatic ductal adenocarcinoma	neutrophils (mice)	tumor promotion	NETs produced by neutrophils	[46]
pancreatic, lung, or skin tumors	neutrophils (mice)	tumor promotion	NETs produced by neutrophils	[47]
breast cancer (cell line)	discrete neutrophil stimulated phorbol 12-myristate 13-acetate (healthy human donors)	tumor promotion	NETs produced by neutrophils	[48]
leukemia (cell line)	NB4 acute pro-myelocytic leukemia cells containing NE gene	tumor promotion	NE produced by acute pro-myelocytic leukemia cells	[49]
thirty-five different human or murine cancers (cell line)	discrete neutrophils (healthy mice and human donors)	tumor suppression	NE produced by neutrophils; CTL-induced tumor killing	[50]
lung carcinoma	discrete neutrophils (healthy mice)	tumor suppression	NE produced by neutrophils; CTL-induced tumor killing	[50]
melanoma	β-glucan stimulated neutrophils (mice)	tumor suppression	trained immunity	[51]
lung carcinoma	neutrophils after immunotherapy (mice and human)	tumor suppression	Interferon gene	[30]
melanoma	neutrophils after immunotherapy (mice)	tumor eradication	iNOS-dependent mechanism	[37]
melanoma and metastatic seeding	neutrophils after combination activated (mice and human)	tumor eradication	ROS-induced tumor killing, CTL-induced tumor killing	[38]

## Data Availability

The datasets for this study can be found in [Pubmed] [https://pubmed.ncbi.nlm.nih.gov/].

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
