# Peer review of "Role of Neutrophils in Anti-Tumor Activity: Characteristics and Mechanisms of Action"

_cancers, 2025, doi:10.3390/cancers17081298_

Round 1
Reviewer 1 Report
Comments and Suggestions for Authors
Three major points for improving this manuscript is the following.
- Figure 1 is oversimplified. More details may make this figure more informative and better complementing the text in section 1. The other figures are also quite simple. Figure 4 is too similar to a figure in this paper, “Targeting myeloid cells with bispecific antibodies as novel immunotherapies of cancer”. (Expert Opin Biol Ther. 2022 Aug;22(8):983-995).
- Section 2 "Neutrophils and tumors" and Section 4 "Phenotypes of Neutrophils in Tumor" can be reorganized and divided into sub-sections to improve the clarity of these sections with dense text. The last section, conclusion and perspective, is also too dense with text. It should be more concise. Some information in this section could be moved to other sections.
- In the 95 references, only about 15 of them were published after 2020. Adding more recent references may enhance the value of this review article.
Reviewer 2 Report
Comments and Suggestions for Authors
This review presents the characteristics of neutrophils that exhibit both anti-tumor and pro-tumoral effects. It also discussed different strategies to use neutrophils for the treatment of tumors.
This is a clear and well-structured review, presented in clear language and supported by informative figures and a useful table.
Minor points:
The authors should add to figure 2, or alternatively to figure 6, the cell surface markers differentially expressed by anti-tumor versus pro-tumor neutrophils, both in human and mouse.
They should discuss recent advances in the use of neutrophil-based drug delivery systems for tumor therapy.
Reviewer 3 Report
Comments and Suggestions for Authors
This review describes the role of neutrophils in the cancer microenvironment. The authors listed the pro- and antitumor effects of neutrophils. Also, some therapeutic approaches have been described in a certain detail. The figures are good and well organized.
Minor concerns
The legend to Figure 4 does not consider the role of CD47/SIRP alpha signaling. Please modify.
The addition of a figure depicting the possible therapeutic approaches targeting neutrophils could help the reader understand the message of the manuscript.
There are plenty of works similar to this one in the literature. In 2025, inserting the word "neutrophils and cancer review 2025" in PUBMED, more than 100 papers have been published (more than 400 in 2024).
Of certain interest is the focus on the antitumor effects of neutrophils instead of the pro-tumor effect due to MDSC and other neutrophil-derived cells.
Overall, the manuscript is good, but research reports on this topic are too many.
Round 2
Reviewer 1 Report
Comments and Suggestions for Authors
The authors addressed the comments of the reviewers.